# Feasibility of Extruded Brewer’s Spent Grain as a Food Ingredient for a Healthy, Safe, and Sustainable Human Diet

**DOI:** 10.3390/foods11101403

**Published:** 2022-05-12

**Authors:** María Belén Gutiérrez-Barrutia, María Dolores del Castillo, Patricia Arcia, Sonia Cozzano

**Affiliations:** 1Departamento de Ingeniería, Universidad Católica del Uruguay, Montevideo 11600, Uruguay; belen.gutierrez@ucu.edu.uy (M.B.G.-B.); parcia@latitud.org.uy (P.A.); 2Instituto de Investigacion en Ciencias de la Alimentacion, CSIC-UAM, Nicolas Cabrera 9, 28049 Madrid, Spain; mdolores.delcastillo@csic.es; 3Latitud-LATU Foundation, Av. Italia 6201, Montevideo 11500, Uruguay

**Keywords:** amino acids, antioxidants, brewer’s spent grain, dietary fiber, extrusion, nutrition security, proteins, sustainability

## Abstract

This study aimed to determine the effect of the extrusion process on the nutritional and bioactive profiles of brewer’s spent grain (BSG), contributing to nutrition security by applying a circular economy concept. Response surface methodology was used to optimize the effect extrusion parameters (moisture content, screw speed, and barrel temperature ) had on BSG’s soluble dietary fiber, free glucose, and overall antioxidant capacity. Proximate composition analyses, amino acid profile, extractable polyphenolic content, and antioxidant capacity of BSG and brewer’s spent grain extruded under optimal conditions (BSGE) were carried out. Food safety was analyzed by their microbiological quality, gluten, and acrylamide content. Optimal extrusion conditions were 15.8% of moisture content, 164.3 revolutions per min and 122.5 °C. BSGE presented 61% more soluble dietary fiber than BSG, lower digestible starch, 0.546% of free glucose, and protein quality parameters mostly like those reported for egg, soy, and milk. Despite this, BSG’s overall antioxidant capacity was not improved after thermomechanical processing; BSGE had significantly higher extractable polyphenolic content in its alkali extracts, which were determined qualitatively by high-performance liquid chromatography quadrupole time-of-flight assay in its hydro-alcoholic acid extracts. Furthermore, although it is not gluten free, BSGE is a safe food ingredient with acceptable microbiological quality and no acrylamide.

## 1. Introduction

Many people worldwide do not have year around access to a safe, affordable, and wholesome diet. In fact, 1 in 3 people suffers from at least one type of malnutrition such as hunger, micronutrient deficiency, obesity, or diet-related noncommunicable chronic diseases [1]. Therefore, nutrition security is defined as “having consistent access, availability and affordability of foods and beverages that promote well-being and prevent (and if needed, treat) disease, may be the next needed approach to inform clinical care and public policy” [2].

Paradoxically, there is an overproduction of food in developed countries creating a significant amount of waste [3]: 129 Mt of food waste along the whole supply chain in the European Union in 2011 [4]. Food waste is a source of pollution generation, so its proper management is essential [5]. Although preventing the generation of waste throughout the food supply chain is the preferred strategy [4], the circular economy addresses the possibility of revaluing the agro-industrial by-products, giving origin to new added-value products with functional compounds that benefit the consumer and industry [3,4,6]. Therefore, it presents an “opportunity to increase food security, foster productivity, and economic efficiency, promote resource and energy conservation, and address climate change” [7]. In line with this, “sustainable diets are diets with low environmental impact, which contribute to food and nutrition security and to healthy life for present and future generations” [8].

In this context, interest arises in revaluing brewer’s spent grain (BSG) as a functional ingredient. BSG accounts for 85% of the brewing industry by-products and 39 million tons are produced worldwide [9,10]. BSG is barely used due to its high moisture content (80%), which makes it difficult to transport and store, making it an unstable product prompt to microbial growth [3,11]. This requires that BSG be stabilized and stored under appropriate conditions after production [9]. Nevertheless, it is widely known that BSG contains significant amounts of valuable components that remain mostly untapped [10]. It is a heterogeneous substance that corresponds to the insoluble grain elements obtained during lautering [9,11]. BSG consists of layers of peel, pericarp, and seeds with residual amounts of endosperm and aleurone from barley [3]. It is a lignocellulosic material, and its major components are dietary fiber (50%) and protein (30%) [9]. Furthermore, “since most of the phenolic compounds of the barley grain are contained in the husk and hydroxycinnamic acids accumulate in the cell walls, BSG is a potentially valuable source of phenolic acids” [12].

Therefore, BSG’s sustainability, nutritional value, and attractive price, associated with a growing demand for initially unusable industrial by-products, make BSG an interesting raw material for human food production [9]. To do this, extrusion technology represents a viable, transferable opportunity with a beneficial impact on the functional, technological, and food safety characteristics of the product.

Extrusion is a thermomechanical process that combines several unit operations such as conveying, kneading, mixing, and forming in a single unit [13]. Food extruders belong to the family of HTST equipment, capable of performing cooking tasks under high pressure [14]. The extrusion process has revolutionized the food industry [13], being able to produce infant cereals, snacks, pasta, or bakery products [14], as well as intermediate products. Apart from the flexibility in its applications, interest in the extrusion effect on physicochemical, functional, and nutritional characteristics of food matrices has increased [15].

The relevance of the extrusion applied to the BSG lies mostly in its effect on fiber. It may improve the soluble dietary fiber and insoluble dietary fiber ratio, as the mechanical stress breaks the polysaccharide bonds, causing the release of oligosaccharides and, therefore, an increase in the soluble fiber content [14]. Furthermore, the shear stress process may release phenolic compounds trapped in dietary fiber [16], which might later have an improvement in their bioaccessibility. Nevertheless, the extrusion process may favor the Maillard reaction, which could negatively cause the reduction in essential amino acids content or favor acrylamide formation, reducing the nutritional value of the food matrix [15,17]. All of these possible changes in the nutritional and functional value of BSG depend on processing conditions, which have to be carefully established to allow new foods and/or functional ingredients to be obtained.

The aim of this study was to determine the effect of the extrusion process on the nutritional profile and bioactive compounds of brewer’s spent grain (BSG), contributing to nutrition security by applying the circular economy concept.

## 2. Materials and Methods

### 2.1. Materials

All chemicals used were of reagent grade. Megazyme (Dublin, Ireland) reagents were used for total dietary fiber (K-TDFR-200A), total starch (K-RSTAR), and free glucose (K-GLUC) content. Folin reagent, 2,20-azinobis-(3-ethylbenzothiazoline-6-sulfonic acid) diammonium salt (ABTS), 6-hydroxy-2,5,7,8-tetramethylch-roman-2-acid (Trolox), fluorescein (FL) disodium salt, 2,20-azobis (2-methylpropionamidine) dihydrochloride (AAPH), gallic acid, and caramel were purchased from Sigma-Aldrich (St. Louis, MO, USA).

### 2.2. Raw Material

Brewer’s spent grain (BSG) was provided by Fábricas Nacionales de Cerveza (Minas, Uruguay) from the AB InBev group. It was obtained from the production of a lager beer made up of barley and maize. Firstly, BSG was dried in a convection oven at 45 °C ± 2 °C until its moisture content was less than 10 g/100 g. Secondly, it was ground in a laboratory mill (Retsch ZM 200) to reach a particle size of 0.5 mm.

### 2.3. Optimization of Extrusion Process

Response surface methodology (RSM) was used to optimize the BSG extrusion process. Experiments were performed according to a central composite rotary design (CCRD) with three independent variables tested at 5 different levels and 5 center points. It resulted in 19 experiments which were performed by duplicate and assay 0 corresponds to unextruded BSG (Table 1). Independent variables used were moisture content (MC), screw speed (SS), and extruder barrel temperature (BT). The assays and the independent variables in their coded and actual levels are shown in Table 1. Independent variables levels were established after preliminary trials and previous works [15,16,17,18,19].

Three response variables were chosen to study the effect of the extrusion process in BSGs:

Soluble dietary fiber (SDF) content was determined following the AOAC 991.43 method as described in Section 2.4.1.

Free glucose (FG) content was determined using a glucose oxidase-peroxidase kit (Megazyme, Dublin, Ireland) [20].

Overall antioxidant capacity (OAC) was estimated by the ABTS Quencher method as proposed by Gökmen et al. [21].

For each of these dependent variables, a second-order polynomial response equation was used to predict its experimental behavior:(1)$$Y_{k}=B_{0}+B_{i}\overset{3}{\underset{i=1}{\sum }}X_{i}+B_{ii}\overset{3}{\underset{i=1}{\sum }}X_{ii}{}^{2}++B_{ij}\overset{3}{\underset{i<j=1}{\sum }}X_{i}X_{j}$$
where $Y_{k}$ with k, between 1 and 3, is the measured response variable,$\;B_{0\;}$ is the intercept, $B_{i}$ are coefficients of the first-order terms, $B_{ii}$ are coefficients of the quadratic terms, $B_{ij}$ are coefficients of the cross product, and $X_{i},\;X_{j}$, with i,j between 1 and 3, are the independent variables. Analyses of the response variables were performed in triplicate and the data were reported as the mean ± standard deviation. Analysis of variance was performed to examine the statistical significance of the model terms. Coefficient of determination ($R^{2}$) and lack of fit test were calculated to establish the adequacy of the mathematical models for each dependent variable.

The optimum extrusion conditions were determined by applying multiple desirability function by maximizing the three response variables. The response surface plots and statistical analysis were performed using Design Expert 11 (State-Ease Inc., Minneapolis, MN, USA) software. Brewer’s spent grain extruded under optimal conditions was named BSGE.

Prior to extrusion, BSG samples were humidified and left standing overnight at room temperature. The extrusion experiments were carried out using a single screw extruder (Brabender Co Corderdo E330). The die diameter, screw diameter, and screw length of extruder were 4 mm, 1.8 cm, and 40 cm, respectively. The extruder barrel had three sections whose temperatures were kept the same and the feeding rate was constant at 100 rpm. The BSG extrudates were cooled at room temperature and grounded using a laboratory mill (Retsch ZM 200) to a particle size of 0.5 mm. Finally, the milled extrudates were kept at −20 °C to stop any enzymatic reaction until further analysis.

### 2.4. Nutritional Characterization

The following analysis was performed on stabilized brewer’s spent grain (BSG) and extruded brewer’s spent grain under optimal conditions (BSGE).

#### 2.4.1. Dietary Fiber Content

Total dietary fiber (TDF), insoluble dietary fiber (IDF), and soluble dietary fiber (SDF) were measured following the AOAC 991.43 enzymatic-gravimetric method [22]. Briefly, 1 g of sample (in duplicate) was mixed with 40 mL of MES-TRIS buffer solution (0.05 M pH 8.2) in a beaker. Samples underwent subsequent enzymatic digestion: heat-stable α-amylase (3000 U/mL, pH 8.2, 30 min, 100 °C), protease (350 tyrosine U/mL, pH 8.2, 30 min, 60 °C), and amyloglucosidase (3300 U/mL, pH 4.1–4.8, 30 min, 60 °C). To obtain IDF residue, the enzymatically digested sample mixtures were filtered using a previously weighed crucible. Solutions of the obtained filtrate were precipitated with 95% of ethanol for 1 h at room temperature. After that, it was filtered using another previously weighed crucible. The obtained residue corresponded to wet SDF. Crucibles containing IDF and SDF residues were dried overnight at 105 °C and weighed afterwards. Protein content of the first residue obtained for IDF and SDF was determined by AOAC 984.13. Ash content was determined by incineration at 525 °C for 5 h of the second residues of IDF and SDF. Protein and ash content was subtracted. TDF was obtained as the sum of IDF and SDF. Experiments were performed in triplicate and results were expressed as a percentage of dry matter.

#### 2.4.2. Starch Content

Total starch, digestible starch, and resistant starch were measured according to AOAC 2002.02 [22].

In screw cap tubes, 100 mg of samples was mixed with 4 mL of a solution containing 10 mg/mL of pancreatic α-amylase and 3 U/mL of amyloglucosidase, dissolved in maleate buffer (100 mM, pH 6.0). Samples were incubated at 37 °C for 16 h under constant agitation, 500 revolutions per minute (rpm), using an Eppendorf ThermoMixer ™ C. After that, 4 mL of ethanol 96% was added and stirred on a vortex mixer. Tubes were centrifuged at 13,000 rpm for 10 min.

For determining non-resistant starch, supernatants were collected in a 100 mL volumetric flask. Pellets were resuspended with 2 mL of 50% ethanol and centrifuged (1300 rpm, 10 min). This step was performed twice, and supernatants were combined in a 100 mL volumetric flask. The volume was adjusted to 100 mL with 100 mM sodium acetate buffer (pH 4.5). Then, 0.1 mL aliquots of this solution were incubated with 10 μL of amyloglucosidase solution (300 U/mL) at 50 °C for 20 min. Finally, 3 mL of glucose oxidase/peroxidase reagent was added, and the tubes were further incubated at 50 °C for 20 min. Glucose content was determined by measuring absorbance at 510 nm.

For determining resistant starch, a magnetic stirrer bar and 2 mL of 2 M KOH were added to each screw cap tube and its pellets were resuspended by stirring for 20 min in an ice/water bath over a magnetic stirrer. Then, 8 mL of sodium acetate buffer (1.2 M, pH 3.8) was added to each tube. Afterwards, 0.1 mL of amyloglucosidase solution (3300 U/mL) was added and incubated at 50 °C for 30 min under constant agitation (500 rpm). Later, tubes were centrifuged at 13,000 rpm for 10 min. Supernatants were used to determine the glucose content by the glucose oxidase method.

Total starch content was determined as the sum of resistant starch and non-resistant starch content.

Experiments were performed in triplicate and the results were expressed as a percentage of dry matter.

#### 2.4.3. Free Glucose Content

Free glucose (FG) was determined following Protonotariou et al. [20]. Briefly, 0.1 g of sample was mixed with 2 mL of aqueous ethanol (80%) for 5 min at 85 °C. Afterwards, the mixture was centrifuged for 10 min at 2000 g and free glucose from the supernatant was determined using a glucose oxidase-peroxidase kit (Megazyme, Dublin, Ireland) using a Shimadzu 1800 UV–Visible spectrophotometer. Analyses were performed in triplicate and the results were expressed as a percentage of dry matter.

#### 2.4.4. Protein Content and Amino Acid Analysis

Protein content was determined by the AOAC 984.13 method [22] and the results were expressed as a percentage of dry matter.

The amino acid (AA) composition was determined using a Biochrom30 series amino acid analyzer (Biochrom Ltd., Cambridge Science Park, Cambridge, UK) according to the methodology proposed by Spackman et al. [23], based on ion-exchange chromatography and post-column derivatization with ninhydrin. Prior to analysis, samples were exposed to acid conditions (HCl 6N) for 22 h at 110 °C. Results were expressed as g of AA/100 g of sample on a dry weight basis. Analyses were performed by Centro de Investigaciones Biológicas (CSIC).

The amino acid profiles of BSG and BSGE were used to evaluate the nutritional quality of their protein content by calculating several parameters as follows:The protein efficiency ratio (PER) was based on the following three equations [24]:
(2)$$PER_{1}=-0.684+0.456\times Leu-0.047\times Pro$$
(3)$$PER_{2}=-0.468+0.454\times Leu-0.105\times Tyr$$
(4)$$PER_{3}=-1.816+0.435\times Met+0.78\times Leu+0.211\times His-0.044\times Tyr$$

Satiety indicators (SAT) [25,26]:


(5)
$${SAT}_{1}\left(\frac{g}{100\;g\;of\;protein}\right)=Leu+Ile+Lys+Thr+Tyr$$



(6)
$${SAT}_{2}\;\left(\frac{g}{100\;g\;of\;protein}\right)=Thr+Tyr$$


The essential amino acid index (EAAI) was calculated as the geometrical mean of the ratio of all the EAA in the evaluated food matrix to their content in a highly nutritive reference protein such as whole egg [27].

(7)$$\%EAAI=100\times \sqrt[{n}]{\frac{His_{a}\left(\frac{g}{100\;g\;of\;protein}\right)\times \ldots \times Val_{a}\left(\frac{g}{100\;g\;of\;protein}\right)}{His_{b}\left(\frac{g}{100\;g\;of\;protein}\right)\times \ldots \times Val_{b}\left(\frac{g}{100\;g\;of\;protein}\right)}}\;$$
where “*a*” corresponds to the test sample, “*b*” to whole egg, and “*n*” to the number of EAA.

The biological value (BV) [27] and nutritional index (NI) [28] were calculated as follows:


(8)
BV=1.09×EAAI−11.7



(9)
$$NI=\frac{EAAI\times Protein\;content\;\left(\%\right)}{100}$$


The amino acid scores were calculated using the next formula [28] and based on the amino acid patterns proposed by the FAO/WHO/UNU for infants [29]:


(10)
$$Amino\;acid\;score\;\left(\%\right)=\frac{Value\;of\;an\;essential\;amino\;acid\;\left(\frac{g}{100g\;of\;protein}\right)}{FAO/WHO\left(1973\right)\;value\;for\;an\;essential\;amino\;acids}\times 100$$


#### 2.4.5. Fat Content

Fat was estimated following the ISO-6492-1999 procedure.

#### 2.4.6. Ash Content

Ash was determined in a muffle furnace following ISO 5984-2002.

### 2.5. Functional Characterization

#### 2.5.1. Antioxidant´s Extraction

Two different extractions were performed to extract the antioxidants present in BSG and BSGE, with varying extraction solvent and time length. A hydro-alcoholic acid extraction (EHAA) was performed according to Fernández et al. [30]. Six screw cup tubes were prepared with 1 g of each sample and 10 mL of the extraction solvent (methanol:water:formic acid, 70:25:5). Three tubes were kept under agitation, 500 revolutions per minute (rpm), at ambient temperature for 5 min, and the remaining three tubes were stirred (500 rpm) for 24 h. Finally, supernatants were recovered after centrifugation at 11,000 rpm for 10 min. Additionally, an aqueous alkaline extraction was made following Stefanello et al. [31] with slight modifications. Six screw cup tubes were prepared with 1 g of each sample and mixed with 10 mL of 0.75% NaOH. Three tubes were left under agitation for 5 min and the remaining three for 24 h. The mixture was centrifuged at 11,000 rpm for 10 min, and the supernatant´s pH was adjusted below 6 with 6 M HCl. A second centrifugation was performed under the same conditions and the supernatant was recovered and kept for analysis. All extractions were performed in triplicate.

#### 2.5.2. Extractable Phenolic Content (EPC)

Extractable phenolic content of EHAA extracts and aqueous alkaline extracts was determined by the Folin–Ciocalteu micromethod according to Iriondo-Dehond et al. [32] with slight modifications. Briefly, 10 µL of extract was added to 150 µL of Folin solution (1:10). After 3 min, 50 µL of sodium carbonate solution (3%) was added and the mixture was incubated for 2 h at 37 °C. Finally, absorbance was measured at 735 nm using a microplate reader (BioTek Epoch 2 Microplate Spectrophotometer, Winooski, VT, USA). A gallic acid calibration curve was used for quantification (0–1 mg/mL). Analyses were performed in triplicate. Results were expressed as mg of gallic acid equivalent (GAE) per gram of sample on a dry weight basis (dwb).

#### 2.5.3. Antioxidant Capacity of the Extractable Compounds

The antioxidant capacity needed to be measured for BSG and BSGE EHAA extracts and aqueous alkaline extracts.

The ABTS radical cation decolorization method described by Re et al. [33] was adapted to the micromethod as follows. ABTS stock solution was prepared by diluting the reagent with distilled water to a concentration of 2.5 mM. Then, 2.5 mL of this solution was mixed with 44 µL of 140 mM potassium persulfate and left standing in the dark for 16 h for the complete formation of ABTS radical cation. For the study, the radical solution was diluted with PBS (5 mM, 7.4) to an absorbance of 0.70 (±0.02) at 734 nm. Then, 270 µL of this solution was added to 30 µL of extract and left standing for 20 min in the dark. Absorbance was read at 734 nm using a microplate reader (BioTek Epoch 2 Microplate Spectrophotometer, Winooski, VT, USA). A Trolox standard curve was used for quantification (0–1 mM). Analyses were performed in triplicate. Results were expressed as mg of Trolox equivalent (TE) per gram of sample in dwb.

Total antioxidant capacity by oxygen radical absorbance capacity (ORAC) was performed according to Ou et al. [34]. Briefly, 25 µL of extracts was mixed with 150 µL of fluorescein (11.12 × 10^−2^ µM) and incubated for 30 min at 37 °C. Then, 25 µL of AAPH (15.3 × 10^−2^ M) was added to start the reaction. Its kinetics was followed for an hour at 37 °C by measuring the fluorescence every minute in a microplate reader (BioTek Cytation5 Cell Imaging Multi-Mode Reader, Winooski, VT, USA) at excitation and emission wavelengths of 485 nm and 530 nm, respectively. A Trolox standard curve was used for quantification (0–150 µM). All measurements were performed in triplicate and the results were expressed as μmol of Trolox equivalent (TE) per gram of sample in dwb.

#### 2.5.4. Analysis of Phenolic Compounds by HPLC-QTOF Assay

HPLC equipment (Agilent 1200) was equipped with a quaternary pump (G1311A), coupled degasser (G1322A), thermostated automatic injector (G1367B), thermostated column module (G1316A), and diode array detector (G1315B). It was coupled to a mass spectrometer (Agilent G6530A Accurate Mass QTOF LC/MS) with an atmospheric pressure electrospray ionization source with JetStream technology. Control software included Masshunter Data Acquisition (B.05.00) and Masshunter Qualitative Analysis (B.07.00).

EHAA extracts of BSG and BSGE and all standards solutions were injected at a volume of 20 µL in a ZORBAX Eclipse XDB-C18 column (150 mm × 4.6 mm × 5 µm) at 40 °C. The solvent systems were 0.1% formic acid (solvent A) and 0.1% formic acid diluted in acetonitrile (solvent B). The elution gradient used was (time, % of solvent A): 0 min, 95; 20 min, 85; 30 min, 70; 35 min, 50, 37 min, 95, 45 min, 90%.

Identification was performed by comparison of the molecular formula, retention time, and previous references. A calibration curve was conducted for ferulic acid (1–16 µg/mL).

This quantification was performed by the Analysis Service Unit facilities of the Institute of Food Science, Technology and Nutrition (ICTAN, CSIC, Madrid, Spain).

#### 2.5.5. Melanoidin Content

Aqueous extraction of melanoidins from BSG and BSGE was performed as described by Patrignani et al. [35] using Amicon^®^ Ultra-15 Centrifugal Filter Units with a 10 kDa nominal molecular mass cutoff membrane. The absorbance of the retentates containing the high molecular weight (HMW) fraction was measured at 360 nm using a microplate reader (BioTek Epoch 2 Microplate Spectrophotometer, Winooski, VT, USA). Caramel was used as a standard. Experiments were performed in triplicate and the results were expressed in mg of caramel melanoidins equivalent per gram of sample in dwb.

### 2.6. Food Safety Analysis

#### 2.6.1. Microbiological Quality

BSG and BSGE were microbiologically analyzed to evaluate the safety of their use as food ingredients. ISO 4833 was followed to obtain the total count of aerobic microorganisms. The total count of molds and yeasts was performed according to ISO 7954. The presence of aerobic endospore was determined as described by Iriondo-DeHond et al. [36]. Results were expressed as colony-forming units (CFU)/g.

#### 2.6.2. Gluten Content

Gluten was determined for BSG and BSGE by the National Center of Biotechnology in Spain according to the enzyme-linked competitive immunoassay R5 method as described in Mena et al. (2019) [37].

#### 2.6.3. Acrylamide Content

The quantification of acrylamide in BSG and BSGE was carried out by Coffee Consulting S.L, which uses approved analytical methods for its detection and quantification according to Commission Regulation (EU) 2017/2158. Samples were subjected to a solid–liquid acrylamide extraction, cleaned afterwards with Carrez, and filtered. Detection was performed by an HPLC-MS/MS method based on ISO 18862:2016 as described by Mastovska et al. (2006) [38].

### 2.7. Statistical Data Analysis

Analyses were performed in triplicate, and all data were reported as the mean ± standard deviation. One-way analysis of variance (ANOVA) was performed on each assay, and differences between samples were determined by the Tukey test (α ≤ 0.05). Analyses were performed using XLSTAT Version 2011 (Addinsoft 1995–2010, Paris, France).

## 3. Results and Discussion

### 3.1. Optimization of Extrusion Process

#### 3.1.1. Statistical Analysis and Model Fitting

Results for soluble dietary fiber (SDF), free glucose (FG), and overall antioxidant capacity (OAC) under different extrusion conditions set by the experimental design are shown in Table 1. These were used for multiple regression analysis to determine the fitting model for each dependent variable. The coefficients’ significance was evaluated by using the *p*-value in all three fitted models. Nonsignificant terms (*p* > 0.05) were excluded, and the regression was refitted respecting hierarchy. Model coefficients in terms of coded factors and their significance are shown in Table 2, while actual equations are as follows:(11)$$\begin{array}{ll} Y_{1}=-42.5818 & +0.9175X_{1}+0.2212X_{2}+0.2880X_{3}-0.0023X_{1}X_{2}\\ & -0.0023X_{1}X_{3}-0.0006X_{2}X_{3}-0.0068X_{1}{}^{2}-0.0003X_{2}{}^{2}\\ & -0.0006X_{3}{}^{2\;} \end{array}$$
(12)$$\begin{array}{ll} Y_{2}=-7.3276+ & 0.1486X_{1}+0.2182X_{2}-0.0546X_{3}-0.0012X_{1}X_{2}\\ & +0.0028X_{1}X_{3}-0.0002X_{2}X_{3}-0.0107X_{1}{}^{2}-0.0006X_{2}{}^{2}\\ & +0.0001{X_{3}}^{2} \end{array}$$
(13)$$\begin{array}{ll} Y_{3}=+3.5752- & 0.0483X_{1}-0.0088X_{2}-0.01725X_{3}+0.0003X_{1}X_{3}\\ & +0.00003X_{2}{}^{2}+0.00004{X_{3}}^{2} \end{array}$$

F-value, lack of fit, coefficient of determination (R^2^), and coefficient of variance (CV) were used to evaluate the adequacy of the fitted models (Table 2). A close correlation between the data and fitted models was achieved as all R^2^ obtained were above 0.9; i.e., only 10% of the variation of the results is not explained by the model equation. Moreover, all models had significant F-value (*p* < 0.05), nonsignificant lack of fit (*p* > 0.05), and coefficients of variance lower than 10%, showing the models had reasonable reproducibility. Hence, the selected models were adequate to describe the effect of extrusion operating parameters (MC, SS, BT) on SDF, FG, and OAC, serving as useful information for predicting the optimal extrusion conditions.

#### 3.1.2. Soluble Dietary Fiber (SDF)

The fitting model for SDF corresponded to a second-order polynomial equation (11), which agreed with the models presented in [39,40]. Nevertheless, linear response surface models for SDF have also been published [41]. The significance and relative impact of each model term are shown in Table 2. All quadratic terms were significant (*p* < 0.05) and had a negative effect on SDF. Indeed, the BT quadratic term was the one with the greatest effect on the response variable. These mathematical trends were consistent with the fitting models of previous works [39,40]. However, there were different results regarding the remaining terms. This study determined that all interactions between independent variables were significant (*p* < 0.05) and negatively affected the response variable, while linear terms were not significant (*p* > 0.05). Huang et al. [39] and Jing et al. [40] concluded that all linear terms were significant but not all of the interaction terms had this quality. In fact, only the interaction between BT and SS was found to have a significant (*p* < 0.05) negative effect [40].

Moreover, the response surface plots (Figure 1) provides a visual tool to explain the relationship between the independent variables studied and SDF. Solubilization of dietary fiber seemed to be dependent on the operating conditions and the analytical method used [42]. Extreme extrusion operating conditions for MC, SS, and BT were not favorable for maximizing the BSG’s soluble dietary fiber content. Thus, the maximum SDF content was obtained at mild operating conditions (17.5% MC, 165.4 revolutions per minute (rpm), and 125.5 °C), which caused a 67% increase in BSG´s SDF content in line with the model predictions. As suggested by Huang et al. [39] and Jing et al. [40], an increase in BT and SS, i.e., shear stress, may cause rapid depolymerization of polysaccharides’ glycosidic bonds, improving the solubility of dietary fiber as water-soluble groups are exposed. However, as the temperature continues to increase, water may evaporate, material may get burnt, and a change in friction may lead to the re-aggregation of small molecules. Moreover, higher SS results in shorter residence time for BSG inside the barrel without enough time for its structural transformation. Concerning moisture content, high levels dissipate the mechanical energy input reducing the energy exerted on the materials, while mild levels can increase the pressure in the extruder barrel, frictional force, and shearing force [39].

#### 3.1.3. Free Glucose (FG)

According to the results presented in Table 1, the extrusion process caused different levels of increase in the free glucose content of BSG. To the best of our knowledge, there is no previous work that predicts by RSM the effect of extrusion operating conditions in free sugars. This study suggests all terms involved in the statistical analysis were significant (*p* < 0.05) to predict the effect of extrusion operating conditions on BSG´s free glucose content (Table 2). According to the fit model (12), there are multiple solutions that maximize or minimize the BSG’s glucose content (Figure 2). At mild screw speeds, high temperatures (>150 °C) and moisture content of about 20% or low temperature (<110 °C) combined with low moisture content (~15%) resulted in higher levels of free glucose in extruded BSG. Therefore, higher levels of free glucose were found under extreme operating conditions, as stated by Martínez [43]. During thermomechanical processing, BSG´s glucose content may increase due to starch hydrolysis [15] or cellulose depolymerization. In fact, results showed insoluble dietary fiber (cellulose) might have been over fragmented under extreme operating conditions, resulting in higher FG levels instead of higher SDF content. Besides, matching our results (Figure 2), previous works have reported lower free sugar levels after extrusion due to the Maillard reaction [15,17], which is favored at low moisture contents (<15%) with high temperatures (>170 °C) [15].

#### 3.1.4. Overall Antioxidant Capacity (OAC)

Extruded BSG´s OAC was measured by ABTS Quencher, and the response surface models obtained are shown in Figure 3. According to its model coefficients (Table 2), BSG´s OAC was mostly affected negatively (*p* < 0.05) by MC and BT linear terms. These trends are in accordance with the model found by Ramos-Enríquez et al. [19] for those values found employing the DPPH method in wheat bran. In addition, previous works have also reported a significant effect for the BT quadratic term [19,44,45]. Consequently, extruded BSG presents the highest antioxidant capacity at lower MC and BT at any SS level. Nevertheless, under optimal extrusion conditions, extruded BSG´s antioxidant capacity has no significant difference (*p* > 0.05) from that corresponding to raw BSG. Previous research showed that high-temperature extrusion was liable to damage and decompose labile phenolics and other bioactive compounds, even though it can improve the accessibility of phenolics [44].

#### 3.1.5. Multiple Response Desirability Optimization of BSG Extrusion Conditions

Optimal extrusion conditions were determined by the multiple response desirability function. After analyzing the previous results, maximizing SDF content was prioritized. Regarding free glucose content, it was decided to maximize it too, as the maximum glucose level achieved still corresponded to a low sugar content and higher glucose levels may have a positive sensory impact. In third place, maximization of total antioxidant capacity was not prioritized bearing in mind that there were no significant differences obtained by extrusion, and that accessibility of phenolic compounds might have been improved anyway.

As a result, the optimal extrusion conditions were 15.8% MC, 164.3 revolutions per minute (rpm), and 122.5 °C. The global optimal desirability was 0.76 while the individual optimal desirability was 0.86, 0.70, and 0.56 for SDF, FG, and OAC, respectively.

Under these extrusion conditions, the model estimated 1.79% (dwb) of SDF, 0.546% (dwb) of glucose, and 1.19 mmolTE/g (dwb). Thus, an increase of 62% in SDF and 355% in the FG content of BSG was predicted. However, a reduction of 8% in OAC measured by ABTS Quencher was expected.

### 3.2. Nutritional Characterization

Proximate composition analyses were performed on raw brewer’s spent grain (BSG) and brewer’s spent grain extruded under optimal conditions (BSGE). Results are shown in Table 3.

As a validation of the response surface optimization model, BSG extruded under optimal conditions showed a significant increase of 61% and 328% in its SDF and FG content, respectively. Thus, BSG and BSGE presented significant differences (*p* < 0.05) mainly in their carbohydrate characterization.

BSG dietary fiber is mainly formed by 12–25% cellulose, 20–25% hemicellulose, and 12–28% lignin [9]. Structural changes caused by the thermomechanical process may disrupt the cell wall matrix leading to smaller and more soluble fragments [42]. As a result, BSGE presented less IDF content and higher SDF content than BSG (*p* < 0.05). The amount of IDF decreased was not equal to the increase in SDF. Hence, IDF transformation may have also caused an increase in glucose-free content or the analytical method used underestimated the SDF content.

In addition, a significant reduction (*p* < 0.05) in total starch and digestible BSG’s starch may also be responsible for the increase in BSG’s glucose-free content. A high shearing force causes more severe disintegration and degradation of starch granules and barrel temperature contributes to increasing the degree of starch gelatinization and thus its degree of degradation [41]. Although previous studies have registered both an increase and decrease in resistant starch after extrusion [43], no significant differences were found in BSG’s resistant starch content after processing.

After dietary fiber, protein is the most abundant macronutrient found in BSG and BSGE (Table 3). Its proteins are mostly hordeins (43%) and glutelins (21.5%) [3].

In line with the results presented in Table 4, previous studies have stated that dominant amino acids (AAs) in BSG are glutamic acid, proline, and leucine [11], and minor amino acids are cysteine and methionine [3]. Although the presence of tryptophan in BSG has been documented [3,9], this AA was not detected during the analysis because it was probably decomposed during acid hydrolysis [46].

After extrusion, BSG presented a significant (*p* < 0.05) reduction in its total content of amino acids (Table 4). In fact, the only AAs whose contents remained unchanged after the thermomechanical process were glutamic acid and leucine. The biological availability of AAs may be affected by heating in the presence of reducing sugars leading to a decrease in AAs availability through Maillard reactions [17]. Besides, melanoidin content for BSGE was 0.127 ± 0.003 mg of caramel melanoidins equivalent/g (dwb), which was significantly higher (*p* < 0.05) than the one registered for BSG (0.081 ± 0.004 mg of caramel melanoidin equivalent/g (dwb)). However, considering that the greatest loss of AA was registered for cysteine (19%) and methionine (10%), extrusion might have also caused the oxidation and desulphurization of sulphur-containing AAs [17]. Nevertheless, the protein digestibility value of BSGE might be higher than nonextruded BSG [15].

“Protein nutritional value is dependent on the quantity, digestibility and availability of essential amino acids” [15]. Despite the negative impact (*p* < 0.05) of extrusion on some of the nutritional protein quality parameters (Table 5), BSGE can still be considered a source of good quality protein and an alternative to animal protein.

BSG and BSGE presented 38% and 39% of essential amino acids (EAA), respectively (Table 5). These contents are similar to animal-based proteins (37% ± 2%) and are higher than the mean EAA content for plant-based proteins (26% ± 2%) [46]. Moreover, the amino acid score for BSG and BSGE’s limiting essential amino acid (methionine) is 90% and 81%, respectively.

In addition, BSG and BSGE have the same levels of essential branched-chain amino acids (BCAAs) (Table 5) as milk (19.46 g/100 g of protein), egg (19.48 g/100 g of protein), or soy (17.41 g/100 g of protein) [47]. BCAAs are recommended for muscle tissue build-up [24] and are important for the prevention of type 2 diabetes [48]. Although after extrusion there was a significant reduction (*p* < 0.05) in BCAAs and aromatic amino acids (AAAs) content, Fischer’s ratio remained statistically the same and was higher than the one for milk (1.95), egg (1.84), and soy (1.64) [47]. Decreases in Fischer’s ratio are associated with increasing severity of hepatic dysfunction and dysfunctions in liver metabolism [49]. Finally, a protein with a lower ratio of lysine to arginine has less lipidemic and atherogenic effects. BSG and BSGE presented a lower Lys/Arg ratio than milk (2.23), but similar to eggs (1.05) and higher than soy (0.73) [47].

BSG and BSGE’s essential amino acid index (EAAI) between 70% and 80% (Table 5) means its proteins are useful [24]. Besides, both have an EAAI higher or equal to wheat bran and white flour [27,28]. Biological value (BV) and nutritional index (NI) are functions of the EAAI. The biological value (BV) is a measure of the percentage of protein that is actually incorporated into the proteins of the human body [27]. BSG presented a BV of 76% (Table 5), corresponding to a good nutritional quality protein [27], but BSGE’s BV, 65% (Table 5), was below these standards. However, both BV more than double the BV recorded by Ijarotimi [28] for wheat flour. On the other hand, the nutritional index (NI) which depends on the protein content of the sample, was not as high as the BV (Table 5).

Protein efficiency ratios (PER) for low-quality proteins are below 1.5, whereas the values above 2, such as the ones obtained for BSG and BSGE (Table 5), are typical of high-quality proteins [24]. BSG and BSGE did not have significant differences (*p* < 0.05) in their PER values as this parameter mainly depends on the leucine content.

Finally, “proteins play a significant role in the regulation of appetite, food intake, body weight and body composition” [26]. Satiety indicators, SAT_1_ and SAT_2_, were calculated. Results showed (Table 5) that BSG would have a more effective hunger control effect than BSGE. Nevertheless, SAT_2_ for BSG and BSGE are higher than those for milk (5.41 g/100 g of protein), egg (6.02 g/100 g of protein), and soy (5.14 g/100 g of protein) [47].

### 3.3. Functional Characterization

Ferulic acid and *p*-coumaric acid are the most abundant hydrocinnamic acids in BSG, which can bind to dietary fiber, especially hemicellulose and lignin [3,9,11]. In agreement with the findings shown in Table 6, previous studies have suggested that BSG’s extracts obtained from organic solvents are within 1.3–1.6 mg GAE/g [50]. However, other authors have recorded higher polyphenols levels (16.2–19.5 mg GAE/g dwb) under similar extraction conditions [3].

Different results were found for extractable phenolic compounds (EPC) and their antioxidant capacity depending on extraction time and solvent used (Table 6). The extraction of polyphenols was higher at longer extraction times (*p* < 0.05). The same trend was found when antioxidant activity was measured by ORAC (*p* < 0.05), which remarks the importance of extraction time on the recovery of antioxidant compounds. Furthermore, the most effective extraction solvent for the measurement of EPC in BSG and BSGE was the aqueous solution of NaOH (*p* < 0.05). This agrees with previous studies, three phenolic acids derived from hydrocinnamates (*p*-coumaric acid, trans-ferulic acid, and sinapic acid) were detected under alkaline hydrolysis [6,31]. Alkaline hydrolysis disrupts the cell wall, dissolving lignin and hemicelluloses allowing the release of unbound ferulic acid and *p*-coumaric acid [6]. Nevertheless, alkali conditions may result in structural simplification which cannot be used for characterization purposes [31]. Moreover, extracts obtained under alkaline hydrolysis presented a higher antioxidant capacity (*p* < 0.05) measured by both ABTS and ORAC. Under alkali conditions, arabinoxylans and arabinoxylo-oligosaccharides as well as bioactive peptides from BSG may be extracted, which are also effective antioxidants and potent radical scavengers [51,52].

Regarding the effects of the extrusion process, extruded BSG presented higher content (*p* < 0.05) of EPC than unextruded BSG under alkaline extraction, but not under EHAA (*p* > 0.05). The same trends were found for the antioxidant capacity measured either by ABTS or ORAC, as it is generally proportional to phenolic content. The extrusion process could make a change in the three-dimensional structure of the fiber molecule causing the release of phenolic compounds that are bonded to the structure [19]. Soison et al. [45] also demonstrated an increase in the ABTS antioxidant activities of extruded sweet potato flours after extrusion, particularly at 10% MC regardless of the SS levels. The increase in antioxidant capacity might be due to the products formed during Maillard reactions [19,53,54,55]. In fact, the antioxidant capacity of the high molecular weight (HMW) fractions obtained during the melanoidin extraction of BSGE was 0.275 ± 0.009 µmol TE/g dwb, which was higher (*p* < 0.05) than the one registered for the HMW fraction of BSG (0.178 ± 0.005 µmol TE/g dwb).

Nevertheless, “colorimetric assays do not provide enough information about structural changes in target compounds” [31]. Tentatively identified compounds by retention time, molar mass and bibliography after HPLC analysis of the EHAA extracts for BSG and BSGE are shown in Table 7. Neither in BSG nor in BSGE were ferulic acid and *p*-coumaric acid the most abundant phenolic compounds in the EHAA extracts. This indicates that these compounds may be mainly bound to the structural matrix, as suggested by Verni [56] who did not identify free forms of ferulic acid or *p*-coumaric acid. Instead, C_9_H_10_O_3_ was the principal phenolic compound detected, followed by C_9_H_10_O_4_; both were identified as possible metabolites of caffeic acid which was not found in the analysis but its presence in BSG was reported by previous works [9,50,56]. As shown in Table 7, in general, extrusion caused a 12% increase in the total phenolic components extracted, owing to an increase in the release of *p*-coumaric acid, 2-(3-hydroxyphenyl) propionic acid, ferulic acid, *p*-coumaric acid glucoside, and benzoic acid, as shown in Table 7. In fact, the quantification of ferulic acid’s content, was significantly higher (*p* < 0.05) in BSGE than in BSG, 0.429 ± 0.010 mg FA/100 g (dwb) and 0.147 ± 0.012 mg FA/100 g (dwb), respectively.

### 3.4. Food Safety Analysis

To establish whether BSG or BSGE are safe food ingredients, allergens, process contaminants such as acrylamide, and microbiological quality must be assessed.

Due to the high moisture content, the linings must be treated fresh after brewing to avoid the multiplication of microorganisms [11]. As shown in Table 8, after the thermomechanical process, the number of aerobic microorganisms, aerobic endospores, and molds and yeasts present in BSG were significantly reduced (*p* < 0.05). Although microbiological analyses were performed on dried BSG, the most effective method for its preservation [3], it exceeded the allowed limits of aerobic microorganisms and molds and yeasts for flours and semolinas based on RD 1286/84 BOE 8/8/84. On the other hand, BSGE can be considered a product with suitable microbiological quality for human feeding.

Although, BSGE cannot be considered a gluten-free or low gluten content ingredient, but its gluten content was lower (*p* < 0.05) than for BSG. Moreover, BSG was free of acrylamide, this processing contaminant not being generated during the extrusion process.

## 4. Conclusions

The extrusion process of brewer’s spent grain was optimized by applying the response surface methodology. Optimal extrusion conditions were 15.8% of moisture content, 164.3 revolutions per minute, and 122.5 °C of barrel temperature. Under these operating parameters, a food ingredient for a healthy, safe, and sustainable human diet was obtained. From a nutritional point of view, extruded brewer’s spent grain presented an increase in soluble dietary fiber content of 61%, low sugar levels, lower digestible starch content, and good quality protein. Regarding its functional value, after extrusion, bioactive compounds were made more accessible. Although it cannot be considered a gluten-free ingredient, extruded BSG is a safe product with acceptable microbiological levels and a lack of acrylamide. Overall, under a circular economy concept, extruded brewer’s spent grain is a new sustainable food ingredient that complies with nutrition security premises: it is widely available, affordable, and may promote human health and wellbeing.

## Figures and Tables

**Figure 1 foods-11-01403-f001:**
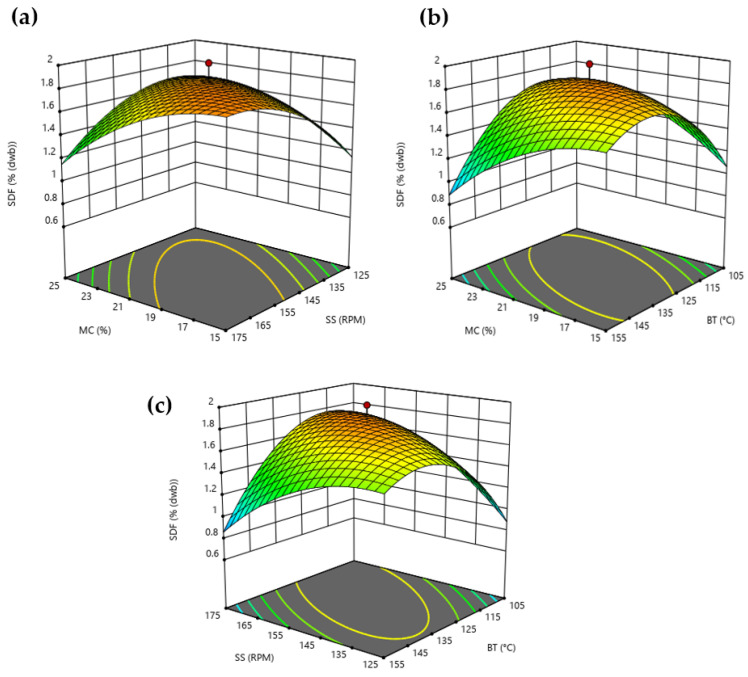
Response surface plots showing the effects of (**a**) moisture content (MC) and screw speed (SS), (**b**) moisture content (MC) and barrel temperature (BT), and (**c**) barrel temperature (BT) and screw speed (SS) on soluble dietary fiber content (SDF).

**Figure 2 foods-11-01403-f002:**
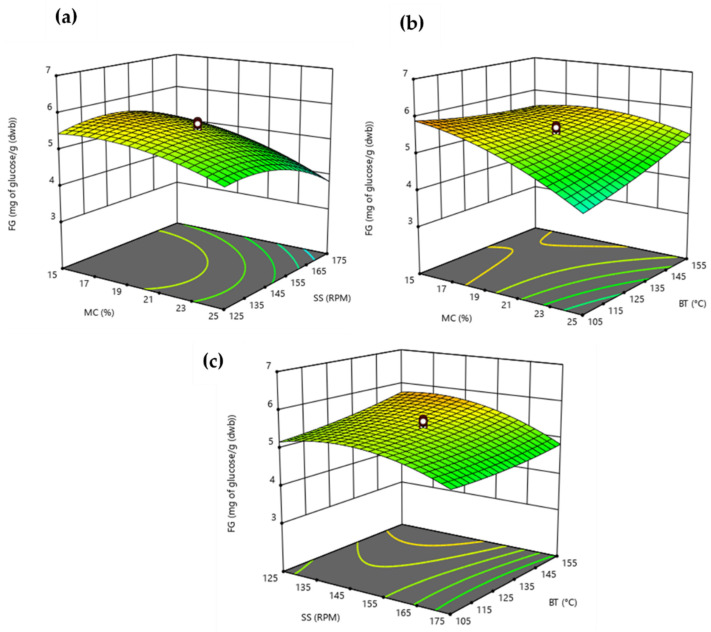
Response surface plots showing the effects of (**a**) moisture content (MC) and screw speed (SS), (**b**) moisture content (MC) and barrel temperature (BT), and (**c**) barrel temperature (BT) and screw speed (SS) on free glucose content (FG).

**Figure 3 foods-11-01403-f003:**
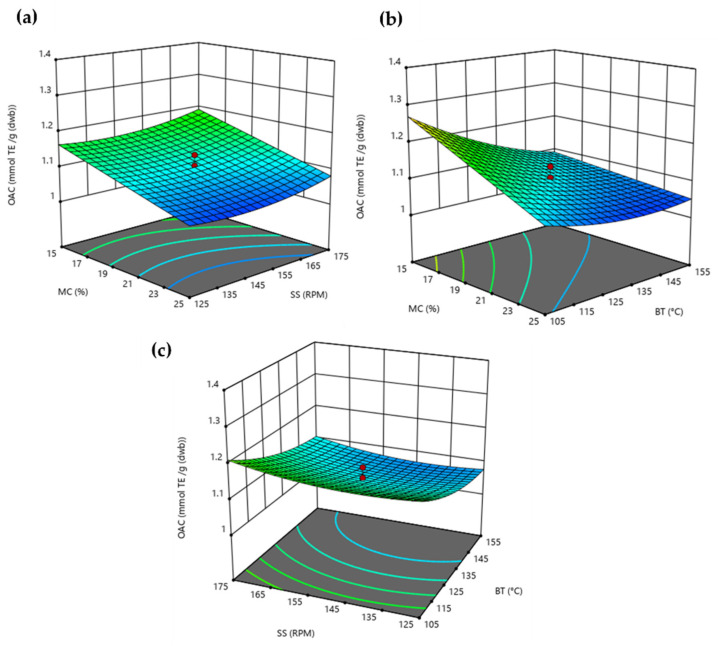
Response surface plots showing the effects of (**a**) moisture content (MC) and screw speed (SS), (**b**) moisture content (MC) and barrel temperature (BT), and (**c**) barrel temperature (BT) and screw speed (SS) on overall antioxidant content (OAC).

**Table 1 foods-11-01403-t001:** CCRD and response values.

Assay	Independent Variables						Dependent Variables		
	Coded			Actual			Responses		
	*X*_1_: MC	*X*_2_: SS	*X*_3_: BT	*X*_1_: MC (%)	*X*_2_: SS (rpm)	*X*_3_: BT (°C)	*Y*_1_: SDF (% dwb)	*Y*_2_: FG (mg glucose/g dwb)	*Y*_3_: OAC (mmol TE/ g dwb)
0	-	-	-	3	0	0	1.103 ± 0.028	1.250 ± 0.028	1.315 ± 0.008
1	−1	−1	−1	15	125	105	1.687 ± 0.064	5.612 ± 0.047	1.276 ± 0.108
2	1	−1	−1	25	125	105	1.235 ± 0.006	5.590 ± 0.032	1.094 ± 0.028
3	−1	1	−1	15	175	105	1.634 ± 0.069	5.525 ± 0.171	1.324 ± 0.110
4	1	1	−1	25	175	105	1.480 ±0.050	5.685 ± 0.132	1.109 ± 0.046
5	−1	−1	1	15	125	155	1.388 ± 0.045	5.584 ± 0.063	1.121 ± 0.070
6	1	−1	1	25	125	155	1.233 ± 0.022	5.537 ± 0.199	1.036 ± 0.038
7	−1	1	1	15	175	155	1.341 ± 0.009	4.857 ± 0.315	1.226 ±0.044
8	1	1	1	25	175	155	1.037 ± 0.012	4.279 ± 0.035	1.101 ± 0.076
9	0	0	0	20	150	130	1.992 ± 0.095	5.826 ± 0.306	1.076 ± 0.030
10	−1.68	0	0	11.59	150	130	1.338 ± 0.030	6.266 ± 0.152	1.180 ± 0.026
11	1.68	0	0	28.41	150	130	1.333 ± 0.055	3.927 ± 0.150	1.027 ± 0.043
12	0	−1.68	0	20	107.95	130	1.229 ± 0.063	4.710 ± 0.181	1.133 ± 0.012
13	0	1.68	0	20	192.05	130	1.255 ± 0.005	6.027 ± 0.204	1.171 ± 0.030
14	0	0	−1.68	20	150	87.96	0.820 ± 0.010	5.451 ± 0.035	1.252 ± 0.018
15	0	0	1.68	20	150	172.05	0.691 ± 0.019	5.649 ± 0.398	1.070 ± 0.054
16	0	0	0	20	150	130	1.775 ± 0.043	5.471 ± 0.265	1.107 ± 0.027
17	0	0	0	20	150	130	1.753 ± 0.084	5.803 ± 0.306	0.991 ± 0.056
18	0	0	0	20	150	130	1.802 ± 0.070	5.595 ± 0.013	1.077 ± 0.072
19	0	0	0	20	150	130	1.738 ± 0.059	5.564 ± 0.266	1.137 ± 0.014

**Table 2 foods-11-01403-t002:** Regression summaries for dependent variables.

Coefficients in Terms of Coded Factors			
Factors	*Y*_1_: SDF	*Y*_2_: FG	*Y*_3_: OAC
Intercept	1.81	5.44	1.10
$\mathrm{MC}\;(X_{1})$	−0.0383	−0.4793 ***	−0.0560 ***
$\mathrm{SS}\;(X_{2})$	+0.0397	−0.3402 ***	0.0145
$\mathrm{BT}\;(X_{3})$	−0.0515	0.1688 ***	−0.0531 ***
$\mathrm{MC}\;\times \;\mathrm{SS}\;(X_{1}X_{2})$	−0.2924 **	−0.1471 ***	ns
$\mathrm{MC}\;\times \;\mathrm{BT}\;(X_{1}X_{3})$	−0.2925 **	0.3468 ***	ns
$\mathrm{SS}\;\times \;\mathrm{BT}\;(X_{2}X_{3})$	−0.3652 ***	−0.1139 **	0.0357 **
$\mathrm{MC}\;\times \;\mathrm{MC}\;(X_{1}{}^{2})$	−0.1688 ***	−0.2679 ***	ns
$\mathrm{SS}\;\times \;\mathrm{SS}\;(X_{2}{}^{2})$	−0.2019 ***	−0.3846 ***	0.0196 *
$\mathrm{BT}\;\times \;\mathrm{BT}\;(X_{3}{}^{2})$	−0.3740 ***	0.0798 **	0.0226 *
**Fit Statistics**			
F-value (model)	20.44 ***	68.12 ***	33.93 **
Lack of fit (*p*-value)	0.5646	0.7886	0.8273
R^2^	0.9633	0.9082	0.9532
Adjusted R^2^	0.9162	0.8948	0.9251
Predicted R^2^	0.7443	0.8743	0.8806
CV	7.07	4.15	2.00

MC, SS, BT, SDF, FG, and OAC represent moisture content, screw speed, barrel temperature, soluble dietary fiber, free glucose, and total antioxidant capacity. * *p* < 0.05; ** *p* < 0.01; *** *p* < 0.001; ns: not significant.

**Table 3 foods-11-01403-t003:** Proximate composition of BSG and BSGE.

	BSG	BSGE
Carbohydrates (g/100 g dwb)		
Total dietary fiber (TDF)	49.66 ± 0.22 ^a^	48.47 ± 0.91 ^a^
Insoluble dietary fiber (IDF)	48.57 ± 0.15 ^b^	46.70± 0.63 ^a^
Soluble dietary fiber (SDF)	1.12 ± 0.03 ^a^	1.77 ± 0.00 ^b^
Total starch	4.498 ± 0.18 ^b^	4.14 ± 0.17 ^a^
Digestible starch	2.68± 0.11 ^b^	2.55 ± 0.06 ^a^
Resistant starch	1.71 ± 0.04 ^a^	1.69 ± 0.01 ^a^
Free glucose	0.13 ±0.00 ^a^	0.54 ± 0.02 ^b^
Proteins (g/100 g dwb)	29.78 ± 0.33 ^a^	29.46 ± 0.11 ^a^
Lipids (g/100 g dwb)	9.61 ± 0.04 ^a^	10.34 ± 0.01 ^b^
Ash (g/100 g dwb)	3.27 ± 0.02 ^a^	3.31 ± 0.06 ^a^

Different letters within the same row mean significant differences (*p* > 0.05).

**Table 4 foods-11-01403-t004:** Amino acid composition of BSG and BSGE.

	Amino Acid (g/100 g dwb)	BSG	BSGE
Nonessential amino acids (NEAA)	Alanine (Ala)	2.054 ± 0.028 ^b^	1.888 ± 0.039 ^a^
	Arginine (Arg)	0.996 ± 0.022 ^b^	0.857 ± 0.020 ^a^
	Aspartic acid (Asp)	1.954 ± 0.0246 ^b^	1.800 ± 0.033 ^a^
	Cysteine (Cys)	0.220 ± 0.017 ^b^	0.176 ± 0.006 ^a^
	Glutamic acid (Glu)	5.360 ± 0.226 ^a^	5.124 ± 0.107 ^a^
	Glycine (Gly)	1.075 ± 0.016 ^b^	0.988 ± 0.020 ^a^
	Proline (Pro)	2.994 ± 0.063 ^b^	2.729 ± 0.083 ^a^
	Serine (Ser)	1.460 ± 0.011 ^b^	1.361 ± 0.030 ^a^
	Tyrosine (Tyr)	0.939 ± 0.020 ^b^	0.768 ± 0.009 ^a^
Essential amino acids (EAA)	Histidine (His)	0.832 ± 0.016 ^b^	0.755 ± 0.017 ^a^
	Isoleucine (Ile)	1.108 ± 0.009 ^b^	1.023 ± 0.020 ^a^
	Leucine (Leu)	3.188 ± 0.150 ^a^	3.003 ± 0.042 ^a^
	Lysine (Lys)	0.931 ± 0.032 ^b^	0.857 ± 0.025 ^a^
	Methionine (Met)	0.589 ± 0.012 ^b^	0.527 ± 0.008 ^a^
	Phenylalanine (Phe)	1.593 ± 0.028 ^b^	1.472 ± 0.030 ^a^
	Threonine (Thr)	1.111 ± 0.011 ^b^	1.014 ± 0.019 ^a^
	Tryptophan (Trp)	n.d.	n.d.
	Valine (Val)	1.453 ± 0.019 ^b^	1.356 ± 0.018 ^a^
Total		27.859 ± 0.264 ^b^	25.698 ± 0.419 ^a^

Different letters within the same row mean significant differences (*p* > 0.05). n.d means not determined.

**Table 5 foods-11-01403-t005:** BSG and BSGE’s nutritional protein quality parameters.

	BSG	BSGE
EAA (g/100 g of protein)	38.786 ± 0.272 ^a^	39.351 ± 0.188 ^b^
Phe + Tyr (AAA) (g/100 g of protein)	8.505 ± 0.151 ^b^	7.605 ± 0.084 ^a^
Ile + Leu + Val (BCAA) (g/100 g of protein)	19.310 ± 0.561 ^b^	18.272 ± 0.243 ^a^
Fischer’s ratio (BCAA/AAA)	2.271 ± 0.092 ^a^	2.403 ± 0.037 ^a^
Lys/Arg ratio	0.935 ± 0.045 ^a^	0.999 ± 0.026 ^a^
%EAAI	76.074 ± 0.261 ^b^	70.698 ± 1.1888 ^a^
BV	71.220 ± 0.285 ^b^	65.361 ± 1.296 ^a^
NI	22.647 ± 0.078 ^b^	21.047 ± 0.359 ^a^
PER_1_	4.671 ± 0.222 ^a^	4.400 ± 0.073 ^a^
PER_2_	4.724 ± 0.224 ^a^	4.434 ± 0.068 ^a^
PER_3_	5.009 ± 0.443 ^a^	4.993 ± 0.098 ^a^
SAT_1_ (g/100 g of protein)	24.443 ± 0.375 ^b^	22.625 ± 0.326 ^a^
SAT_2_ (g/100 g of protein)	6.885 ± 0.102 ^b^	6.050 ± 0.071 ^a^

Essential amino acids (EAA), phenylalanine (Phe), tyrosine (Tyr), isoleucine (Ile), leucine (Leu), valine (Val), branched-chain amino acids (BCAA), aromatic amino acids (AAA), lysine (Lys), arginine (Arg), essential amino acid index (EAAI), biological value (BV), nutritional index (NI), protein efficiency ratio (PER), satiety indicators (SAT). Different letters within the same row mean significant differences (*p* > 0.05).

**Table 6 foods-11-01403-t006:** Phenolic content and antioxidant capacity of BSG and BSGE’s extracts.

Sample	Extraction Method	EPC (mg GAE/g dwb)	ABTS Method (µmol TE/g dwb)	ORAC Method (µmol TE/g dwb)
BSG	EHAA 5 min	1.403 ± 0.056 ^b^	27.126 ± 1.287 ^a^	25.475 ± 1.305 ^a^
	EHAA 24 h	1.569 ± 0.038 ^c^	38.548 ± 0.851 ^a^^,b^	50.188 ± 0.873 ^b^
	Alkaline 5 min	1.587 ± 0.056 ^c^	38.945 ± 0.651 ^c^	216.232 ± 7.951 ^c^
	Alkaline 24 h	3.623 ± 0.124 ^e^	79.143 ± 2.834 ^e^	304.923 ± 5.752 ^d^
BSGE	EHAA 5 min	1.206 ± 0.044 ^a^	26.904 ± 0.906 ^a^	23.337 ± 1.395 ^a^
	EHAA 24 h	1.623 ± 0.088 ^c^	32.084 ± 1.590 ^b^	67.591 ± 0.037 ^b^
	Alkaline 5 min	2.366 ± 0.110 ^d^	52.200 ± 1.215 ^d^	311.769 ± 6.654 ^d^
	Alkaline 24 h	4.221 ± 0.114 ^f^	92.955 ± 4.775 ^f^	492.470 ± 8.691 ^e^

Extractable phenolic content (EPC), gallic acid equivalent (GAE), hydro-alcoholic acid extraction (EHAA), 2,2′-azino-bis (3-ethylbenzothiazoline-6-sulfonic acid) (ABTS), total antioxidant capacity by oxygen radical absorbance capacity (ORAC), Trolox equivalent (TE). Different letters within the same column mean significant differences (*p* > 0.05).

**Table 7 foods-11-01403-t007:** Qualitative analysis of phenolic compounds present in BSG and BSGE.

Proposed Compound	Molecular Formula	Molar Mass (g/mol)	Retention Time (min)	Relative Percentage in BSG * (%)	Relative Percentage in BSGE * (%)	Variation after Extrusion ** (%)
*p*-coumaric acid	C_9_H_8_O_3_	163.0	16.4	0.4	0.9	111.1
2-(3-hydroxyphenyl) propionic acid	C_9_H_10_O_3_	165.1	16.7	51.6	58.8	27.5
Ferulic acid	C_15_H_18_O_8_	193.1	20.0	0.2	0.3	72.4
*p*-coumaric acid glucoside	C_15_H_18_O_8_	325.1	18.2	1.3	1.5	23.9
Hydrodiferulic	C_20_H_18_O_8_	385.1	6.7	7.5	6.2	−7.5
Dihydrocaffeic acid	C_9_H_10_O_4_	181.1	6.7	24.9	19.4	−12.7
Dihydrobenzoic acid	C_7_H_6_O_4_	153.0	4.3	4.0	3.0	−15.6
Hydroxybenzoic acid	C_7_H_6_O_3_	137.0	7.6	5.4	4.5	−5.2
Benzoic acid	C_7_H_6_O_2_	121.0	11.5	4.6	5.7	31.3

(*) Calculated as the ration between each compound chromatogram area and the sum of all areas registered. (**) Calculated as the ratio between the difference of area in BSGE and BSG and its chromatogram area in BSG for each compound.

**Table 8 foods-11-01403-t008:** Food security evaluation of BSG and BSGE.

	BSG	BSGE
Total aerobic count (cfu/g)	(5.150 ± 1.626) × 10^6 b^	(6.550 ± 0.636) × 10^2 a^
Total aerobic endospores count (cfu/g)	(3.250 ± 0.495) × 10^4 b^	6.000 ± 1.414 ^a^
Total molds and yeasts count (cfu/g)	(1.700 ± 0.282) × 10^6 a^	n.d.
Gluten (ppm)	(30.519 ± 2.361) × 10^5 b^	(25.258 ± 1.413) × 10^5 a^
Acrilamida	<20 µg/kg ^a^	<20 µg/kg ^a^

Different letters within the same row mean significant differences (*p* > 0.05). n.d means not determined.

## Data Availability

Data is contained within the article.

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
