# Peer review of "Feasibility of Extruded Brewer’s Spent Grain as a Food Ingredient for a Healthy, Safe, and Sustainable Human Diet"

_foods, 2022, doi:10.3390/foods11101403_

Round 1
Reviewer 1 Report
In my opinion, the manuscript entitled Fesability of the extruded brewers spent grain as a food ingredient for a healthy, safe and sustainable human diet by Gutiérrez-Barrutia et al., is a very interesting article. The introduction is well done and with corelated with the current state of the art, the materials and methods are well described and the results are well discussed and justified.
I have some suggestions, comments, as follows:
1. line 25. Please mention the abbreviations for HPLC-QTOF.
2. Line 121. Please better describe the soluble dietary fiber method here or at lines 152-153.
3. Line 122. There is an extra space before the reference.
4. Lines 156, 273, 276. Please better describe the used methods.
5. Line 388, table 3. I think that letters should be in superscript and in all table the values should have 2 values after point. For instance 49.66 ± 0.21.
6. Regarding the antioxidant activity, there are several studies which mentioned that during extrusion the antioxidant activity could increase, mainly due to Maillard reaction which could improve antioxidant activity (Pasqualone et al., 2020, Igual et al.,2021). How authors better justified their results?
Author Response
Please, see document attached.

Reviewer 2 Report
Dear Editor,
the manuscript describes the effect of extrusion process on the nutritional and functional properties of brewers spent grain, the major by-product of the beer-brewing industry. The authors optimized extrusion processing parameters using response surface methodology and provided an overall characterization of the extruded snack. The experimental design looks adequate and thorough. The methods and data collected are sufficient to draw the appropriate conclusions.
I believe moderate English changes are required.
Author Response
Dear Reviewer,
Thank you very much for your recommendations on our manuscript. Grammar and spelling have been checked and improved all through the manuscript. Changes appear in red.
Best wishes,
María Belén Gutiérrez
Reviewer 3 Report
I would recommend having a native speaker help with the English editing.
Line 32: I would recommend saying year around instead of all the year.
Line 153, 157, 164, 215: experiments were done in triplicate. not by.
Line 199 - 200, 202 - 203. Was BSG held under agitation for 5 minutes and then again at 24 hrs. Or was BSG (5 minutes) and BSGE 24 hrs. It's somewhat confusing.
Line 219: wording. measure, needs to be measured and to should be changed to the word for.
Figure 1: is difficult to read with so many together. Can you separate them to improve the readability of them.
Author Response
Please see document attached.

Reviewer 4 Report
Manuscript Number: foods-1686052
Title: Feasibility of the extruded brewers spent grain as a food ingredient for a healthy, safe and sustainable human diet
Overview and general recommendation
The article structure is compact, sequential and logical. The data are adequate to support the conclusion. The methods section provides sufficient information on sampling, definitions, data collection and data analysis. References are up-dated adequate and correctly cited.
In this study the authors evaluated the effect of extrusion process on the nutritional and bioactive profiles of brewers spent grain.
Minor comments:
- The part of Abstract: In this part I recommend not to use abbreviations.
- What does RPM or HPLC-QTOF abbreviation?
- The part of Materials and methods: This section provides sufficient information on sampling, definitions, data collection and data analysis. Why the milled extrudates were kept at -20°C for further analysis?
- The part of Results and Discussions:
- Line 287: must be explain the abbreviations: SDF, FG and OAC
- Table 2: must be explain the abbreviation: SDF
- Line 326: must be explain the abbreviation: RPM
- Line 350: must be explain the abbreviation: IDF
- Line 376: must be explain the abbreviation: rpm
- Figure 1: must be explain the abbreviations: SDF, RPM, etc.
- Table 5: must be explain all abbreviations in the first column
- Lines 434, 459,464: must be explain the abbreviations: EAA, PER, SAT1, SAT2
- Table 6: must be explain all abbreviations in the table
- Conclusion: Just like the abstract there should be no more abbreviations in this part.
The authors carry out an interesting work and I recommend it for publication after minor revision.
Author Response
Please see document attached.
